# Impact of the COVID-19 Lockdown Measures on Noise Levels in Urban Areas—A Pre/during Comparison of Long-Term Sound Pressure Measurements in the Ruhr Area, Germany

**DOI:** 10.3390/ijerph18094653

**Published:** 2021-04-27

**Authors:** Jonas Hornberg, Timo Haselhoff, Bryce T. Lawrence, Jonas L. Fischer, Salman Ahmed, Dietwald Gruehn, Susanne Moebus

**Affiliations:** 1Institute for Urban Public Health, University Hospital Essen, University Duisburg-Essen, Hufelandstraße 55, 45147 Essen, Germany; timo.haselhoff@uk-essen.de (T.H.); jonas.fischer@uk-essen.de (J.L.F.); salman.ahmed@uk-essen.de (S.A.); susanne.moebus@uk-essen.de (S.M.); 2Department of Landscape Ecology and Landscape Planning, School of Spatial Planning, TU Dortmund University, August-Schmidt-Straße 10, 44227 Dortmund, Germany; bryce.lawrence@tu-dortmund.de (B.T.L.); dietwald.gruehn@tu-dortmund.de (D.G.)

**Keywords:** COVID-19, lockdown, noise, soundscape, built environment, health, mobility

## Abstract

Background: A major source of noise pollution is traffic. In Germany, the SARS-CoV-2 lockdown caused a substantial decrease in mobility, possibly affecting noise levels. The aim is to analyze the effects of the lockdown measures on noise levels in the densely populated Ruhr Area. We focus on the analysis of noise levels before and during lockdown considering different land use types, weekdays, and time of day. Methods: We used data from 22 automatic sound devices of the SALVE (Acoustic Quality and Health in Urban Environments) project, running since 2019 in Bochum, Germany. We performed a pre/during lockdown comparison of A-weighted equivalent continuous sound pressure levels. The study period includes five weeks before and five weeks during the SARS-CoV-2 induced administrative lockdown measures starting on 16 March 2020. We stratified our data by land use category (LUC), days of the week, and daytime. Results: We observed highest noise levels pre-lockdown in the ‘main street’ and ‘commercial areas’ (68.4 ± 6.7 dB resp. 61.0 ± 8.0 dB), while in ‘urban forests’ they were lowest (50.9 ± 6.6 dB). A distinct mean overall noise reduction of 5.1 dB took place, with noise reductions occurring in each LUC. However, the magnitude of noise levels differed considerably between the categories. Weakest noise reductions were found in the ‘main street’ (3.9 dB), and strongest in the ‘urban forest’, ‘green space’, and ‘residential area’ (5.9 dB each). Conclusions: Our results are in line with studies from European cities. Strikingly, all studies report noise reductions of about 5 dB. Aiming at a transformation to a health-promoting urban mobility can be a promising approach to mitigating health risks of noise in cities. Overall, the experiences currently generated by the pandemic offer data for best practices and policies for the development of healthy urban transportation—the effects of a lower traffic and more tranquil world were experienced firsthand by people during this time.

## 1. Introduction

The World Health Organization (WHO) declared the global spread of SARS-CoV-2 causing the coronavirus disease (COVID-19) a pandemic on 11 March 2020 [1]. Without immunization tools, physical distancing is one of the most promising public health measure to slow down the transmission of SARS-CoV-2 [2,3]. Accordingly, many governments reacted to the rapid spread of the virus with the implementation of politically enforced measures, which included economic shutdown, closing of borders, and social and travel restrictions. For instance, in Germany, hard lockdown measures resolved around mid-March 2020 and included the closure of schools, restaurants, shops, leisure facilities, and sports clubs, as well as restrictions on visits to hospitals and nursing care facilities [4]. Additionally, social contacts were restricted, allowing only a limited number of private meetings. One of the most impressive effects of these measures is the reduced mobility of an entire society. 

Despite the dire consequences of the pandemic, however, the deeply profound measures happened to allow for natural experiments that would otherwise simply not be feasible. One of these options is the possibility to measure the effects of reduced mobility, in particular road, railway, and air traffic, on noise levels in urban areas. As environmental noise is one of the most important environmental health risks, with many people being affected [5,6], noise management and mitigation practices are a significant concern for citizens and public health and a key objective for government policy. In order to achieve the transition to sustainable mobility, which is crucial according to the 2030 Agenda for Sustainable Development (UN 2015), the impact of reduced internal-combustion-engine-based mobility could be valuable to project the implications of transformation solutions. 

In Germany, the reduction of economic, human, and social activity has been reflected in a reduced volume of car, rail, and air traffic [7,8,9], which, in turn, changed the perception of the environment of cities and landscapes. One of the most striking visual impressions was of empty streets and highways, which are usually characterized by a high traffic volume. The visual impressions are also underlined by changes in environmental sound. For example, an online survey from Lyon, France, including more than 3000 people, revealed that perceived noise has decreased in both urban and countryside areas [10]. In particular, respondents reported hearing less traffic noise and more nature sounds. Elsewhere, hand-clapping events took place during the coronavirus pandemic, and out-of-order church bell ringing has been observed, e.g., in Australia, which provided unique elements to the soundscape [11,12]; however, no special public sound events were performed by the citizens of Bochum.

First studies using recorded sound pressure levels could demonstrate on average a decrease of 5 dB(A), reflecting the generally perceived noise reduction [10,13,14,15,16]. However, most studies published so far are based on spatially limited data or a small number of monitoring stations. For instance, one study focused on traffic related noise [13], two on public places [15] and city centers with each only one monitoring station [17]. Only three studies so far used a broader representation of the city, including residential areas [10,14,16]. However, it is unknown if these survey sites are selected as typical land use types. Furthermore, since the time and extent of societal lockdown measures differ distinctly between countries and even between cities in one country, further studies are warranted demonstrating environmental effects of this tremendous reduction in traffic volume, which would likely never be observed in normal times.

The aim of our study is to describe the effects of the societal lockdown in Germany on noise levels in the city of Bochum, located in the densely populated and highly trafficked metropolitan Ruhr Area. Here, we focus on the analysis of noise levels before and during lockdown measures considering (i) different land use types in Bochum, (ii) weekdays and weekends, and (iii) time of day.

## 2. Methods

### 2.1. Study Design

For our analyses, we used the comprehensive data sets of the SALVE (Acoustic Quality and Health in Urban Environments) project [18]. The SALVE study design has currently been described in detail [19]. The aim of SALVE is to measure spatial–temporal differences in acoustic environments, using manual and automatic devices at more than 700 different urban locations. The study has been running since spring 2019 in the city of Bochum. Our automated recording started on May 6, thereby including recordings before and during the COVIDa-19 lockdown period in Germany. Bochum is located in the Ruhr Area, populated by approximately 5.1 million inhabitants, thus ranking as the largest urban agglomeration in Germany and the fifth largest in Europe [20]. The acoustic environment of Bochum can be considered as typical urban, i.e., mainly influenced by traffic sounds. 

We divided our observation period into a pre-lockdown and during lockdown phase. We set the crucial date for our analyses to 16 March, the imposition date of the main lockdown measures (Figure A1, Appendix A). The Ministry of Labour, Health, and Social Affairs of North Rhine-Westphalia implemented from 16 March entry bans for travelers returning from risk areas for community facilities, health care facilities, and schools for a period of 14 days after their stay and bans or restrictive limitations on visits in health facilities. Additionally, on 17 March, the closure of bars, clubs, discos, theaters, cinemas and museums, gyms, swimming pools, saunas, adult education centers, music schools, sports clubs, other sports and recreational facilities, arcades, gambling halls, prostitution establishments, and prohibition of all public events was implemented [4]. In the same way, on March 23, further regulation came into force, including restrictions on meetings and gatherings in public of more than two persons with exceptions for relatives, spouses or life partners, household members, minors, compulsory meetings for business, or professional, official, examination, or support reasons [21]. In addition to the measures enforced by law, further protection measures, like recommendations to reduce social contacts and to stay and work from home when possible, were published, particularly by the Robert Koch Institute (RKI) [22]. The RKI is the government’s most important body for the safeguarding of public health in Germany (www.rki.de/EN/Home/homepage_node.html, accessed on 22 April 2021). However, as these were recommendations, travel to work was still permissible (Figure A2 of Appendix A).

Summarizing, we set the lockdown time to five weeks, starting from the first day of the lockdown measures in Germany; accordingly, the last measurement day was 19 April. Hence, we define the pre-lockdown phase five weeks before start of lockdown, which corresponds to 10 February until 15 March 2020. 

### 2.2. Audio Recordings and Land Use Types

For audio recording, we mounted 24 Wildlife Acoustics SM4 Acoustic Recorders on trees on both public and private grounds at a height of about 1.5 m (Figure 1). The distribution of our audio devices in the city of Bochum is mapped in Figure A1 (Appendix A). We excluded two devices from analysis due to technical failure during the specified observation period. Overall, we sampled mono-recordings, saved as .wav (waveform audio) files, at 44,100 Hz with a bit-depth of 16. The devices recorded three-minute samples every 26 min corresponding to 50 recordings per day. The number of all recordings during our observation period totals to n = 76,906. 

We defined the built environment by land use types (LUT) provided by the Regional Association for the Ruhr Area [23]. To better illustrate the development of sound levels before and during the SARS-CoV-2 pandemic in relation to the main sound sources, we defined and categorized new land use types based on LUT, photographs, and assessments of the respective recording sites. The selection was done separately by each member of our team. For example, we grouped mixed forests and deciduous forests as category ‘urban forest’ and ‘residential area’ with different building heights as category ‘residential area’. We resolved disagreements through discussion between the team members. Most inconsistencies occurred between the ‘residential area’ and ‘residential street’; in these cases, we assigned the locations to the original LUT. We also discussed ‘urban agriculture’ more often. This might be due to the special feature of the polycentric Ruhr Area that covers almost 40% of agricultural land, which results in these areas being often on the fringes of residential areas [24]. We classified one site as the ‘main street’ despite this site belonging to the ‘commercial area’ category according to the LUT of the RVR. The decision was reached because this site is located directly on the border between the LUT ‘commercial area’ and ‘main street’; after examining photos and google maps, the main effect on noise levels is very likely caused by the main street with a tram line. Figure 1 depicts the land use categories, one photo as an example, and the number of respective monitoring stations per land use category.

### 2.3. Statistical Analyses

Ahead of our analyses, we checked our data for plausibility. We investigated the sound pressure levels for outliers, checking manually for definitive methodical failures that clearly distort the sound analyses. In these cases, we excluded parts of the dataset accordingly. 

We performed descriptive analyses calculating means and respective standard deviations. Differences between the pre- and during lockdown phases were calculated as Δ = $\bar{x_{d}}$ − $\bar{x_{p}}$, where $\bar{x_{d}}$ represents the mean of the during lockdown phase and $\bar{x_{p}}$ the mean of pre-lockdown phase. As the sound pressure index, we calculated the A-weighted mean dB values of the equivalent continuous sound level L_Aeq_, using the software Kaleidoscope Pro [25]. We calculated the mean L_Aeq_ of all devices by day (referred to as L_Aeq,24h_). Using the L_Aeq,24h_ we describe the mean course of the noise levels for each day over the observation period as well as patterns of specific weekdays. Calculating the mean L_Aeq_ of all devices for each of the two five-week periods pre- and during lockdown (referred to as LA_eq,35d_), we conduct this pre–during comparison. To compare weekdays with weekends, we calculated the mean L_Aeq_ of all devices separately for weekdays (Monday to Friday), Saturdays, and Sundays (referred to as L_Aeq,week_). Lastly, we calculated the daily mean L_Aeq_ per hour for all devices (referred to as LA_eg,1h_) to analyze specific daytime differences. Unless otherwise noted, we performed statistical analyses using IBM SPSS Statistics 27.0.0.0.

## 3. Results

### 3.1. Development of Noise Levels (L_Aeq_,_24h_)

Figure 2 presents the overall daily average course of the L_Aeq,24h_ during the observation period. Despite a high variance of the recordings, the course depicts a clear drop of the mean overall L_Aeq_, interestingly already starting before the administrative resolve. This early drop of the noise levels might be explained by press releases and intensive discussions on social media and TV about upcoming lockdown measures as already performed in other countries, i.e., in China. Furthermore, the health department of the city of Bochum recommended contact reductions as early as on 26 February 2020. Concerns of being infected and subsequent voluntary measures such as staying at home or using a home office started to reduced mobility prior administrative measures (s. Figure A2 of Appendix A). This can accordingly help to explain the downward trend around February 24. The peaks on Sundays on 16 and 23 February are thus not repeated afterwards, although smaller increases can still be seen, but they no longer happen on weekends either. This is possibly due to the fact that people have already reduced their leisure activities, which they otherwise pursued mainly on weekends. One exception is the week before the administrative lockdown showing a distinct increase in dB level. We assume that due to a lot of discussion in the media about suspected supply shortages, many people rushed to do final errands, which might have caused more noise. 

During the lockdown period, the fluctuation of noise levels between the days is not as pronounced as in the course of the pre-lockdown phase. A noticeable decrease of the noise level is observable between 21 March and 22 March, which could be linked to additional administrative restrictions on social contacts resolved on that day. The marked decline on April 10 is probably related to the Easter holidays, on which there were even more reduced office hours (Figure A2 of Appendix A).

The overall decreasing noise level during lockdown shown in Figure 2 is also observable in all land use categories (Figure 3), however on distinctly differing levels. As expected, we found the highest noise levels pre-lockdown in the main streets and commercial areas (68.4 dB resp. 61.0 dB), while in urban forests (50.9 dB) they were lowest (Table 1). During lockdown, the ‘main street’ still shows the highest noise levels with 64.5 dB, whereas the ‘commercial area’ and ‘residential street’ now rank equally, with distinct lower noise levels of 55.9 dB and 55.6 dB compared to the ‘main street’. The ‘urban forest’ turned out to further be the quietest area with a reasonably low noise level of 45.0 dB.

### 3.2. Differences of Noise Levels Pre- and during Lockdown

The observed decrease of noise levels during our study period translates to a mean overall noise reduction of 5.1 dB, corresponding to a mean noise level (L_Aeq,35d_) of 58.0 dB pre-lockdown and 52.9 dB during lockdown (Table 1). Although the decrease turned out to be relatively homogenous through all land use categories, we observed interesting differences. One of the greatest decreases of noise levels occurred in the already comparably quiet ‘urban forest’ with an average reduction of 5.9 dB. We observed the same degree of noise level reduction (5.9 dB) in both ‘green space’ and the ‘residential area’, however the former had an average noise level during lockdown of 50.3 dB and the latter of a remarkable low 49.3 dB. On the other hand, the ‘main street’ showed the lowest noise level decrease of 3.9 dB, dropping from the highest 68.4 dB to 64.5 dB, followed by the ‘small garden near house’ with a decrease of 4.3 dB, however showing an already rather low mean noise level of 51.5 dB pre-lockdown. 

### 3.3. Development of Weekly Noise Levels

Surprisingly, the pre-lockdown period reveals almost no differences of the overall mean noise levels between Sundays (58.0 dB) and weekdays (58.3 dB), whereas Saturdays were found to be the quietest days of the week (56.6 dB). In contrast, a different pattern emerges during lockdown, with Sundays being the quietest day of the week with low 51.5 dB (SD ± 8.3), followed by Saturdays (52.8 dB) and weekdays (53.2 dB). Overall, strongest reductions of the overall noise level between pre- and during lockdown are measured on Sundays (6.5 dB) and the weakest on Saturdays (3.8 dB).

Figure 4 depicts the mean noise levels of the pre- and during lockdown observation period by weekdays and land use category. Most striking in this analysis are the areas ‘urban forest’ and ‘urban agricultural land’ in which the noise levels of weekdays, Saturdays, and Sundays almost converge during the lockdown, whereas pre-lockdown Sundays were loudest in these areas. Overall, in all land use categories, we observe noise levels converging to similar noise levels comparing the pre- and during lockdown phase, however mostly with minimally lower noise levels on Sundays.

### 3.4. Development of Noise Levels by Time of Day

The overall course of mean noise levels of the pre- and during lockdown observation period by hourly time of day (L_Aeq,1h_) is presented in Figure 5. Although the overall patterns pre- and during lockdown are strikingly similar, the pre-lockdown noise level is distinctly higher compared to during lockdown with one exception around 6:00 a.m., when differences are smallest. According to Table 2, we measured the weakest noise level reductions between 5:00 a.m. and 7:00 a.m. (ranging from 1.7–3.7 dB) and highest between 9:00 p.m. (7.7 dB) and 3:00 p.m (6.9 dB).

Looking at the 24-h course for each land use category (Figure 6), it is noticeable that the comparison of noise levels pre- and during lockdown in the ‘main street’ area shows the smallest noise reductions throughout the day. In this area, we also observed the highest diurnal noise reduction at nighttime between 12:00 and 4:00 a.m., however this was in both observation periods, probably due to the resting hours of the tram in this time period.

## 4. Discussion

The aim of the present study is to analyze the effects of the lockdown in Germany on noise levels in a densely populated urban area, focusing on the analysis of noise levels before and during lockdown considering different land use types, weekdays, and time of day. We performed automated sound recordings at 22 locations selected based on different land use types. Our results reveal a distinct overall noise level reduction amounting to 5.1 dB, comparing mean daily L_Aeq_ levels for five weeks before and five weeks during the SARS-CoV-2 induced administrative lockdown measures. We observed noise reductions in each land use category; however, the magnitude of the sound levels differed considerably between the categories. By far the least noise reductions were found in the category ‘main street’ with 3.9 dB, and the greatest reductions were found in ‘urban forest’, ‘green space’, and ‘residential area’, 5.9 dB each. Additionally, we observed that the circadian rhythm of the noise levels during 24 h did not generally differ between the pre- and during noise level courses. Only the noise level differed, owing to more noise in the pre-lockdown period.

Comparing previous studies from different European cities investigating the effects of lockdown measures on noise levels, it is striking that all studies report noise reductions in a very similar order of magnitude of 5 dB(A). Thus, our result fits well into this surprising observation. For instance, Bruitparif (2020) compared a 15-week lockdown lasting from 16 March 2020, to 28 June 2020, with a 14-month pre-lockdown reference period lasting from 1 January 2019, to 29 February 2020, in Paris, France [13]. They report mean noise reductions, indicated as L_DEN_, near high-speed roads and railways of 5.9 dB(A) and 5.3 dB(A). Noise reductions in neighborhoods turned out to be even higher, ranging between 6–20 dB(A). The “Observatoire de l’environnement sonore de la Métropole de Lyon” reports results of 21 recording stations in five French cities (Lyon, Marseille, Grenoble, Saint Etienne, and Toulouse). They compared recordings 54 days during lockdown (17 March 2020, to 10 May 2020) and 69 days during a reference period lasting from 6 January 2020, to March 13, 2020 [10]. They observed a noise reduction of L_DEN_ ranging on average between 4 and 6 dB(A); however, detailed descriptions of the time frame and the types of the selected locations are not provided. Data of 31 recording stations provided by the city council of Madrid, Spain revealed an average L_DEN_ reduction of around 5 dB(A) during the lockdown period (76 days from 16 March 2020, to 31 May 2020) compared to the reference period (38 days from 1 February 2020, to 10 March 2020) [14]. Recordings from London, Great Britain, showed an average L_Aeq_ reduction of 5.4 dB at 11 measurement points during lockdown in spring 2020 compared to recordings at the same locations from spring 2019 [15]. They also report a high spatial variance of L_Aeq_ levels ranging between 1 to 10 dB. In Dublin, Ireland, Basu et al. (2020) [16] measured noise levels at 12 monitoring stations across the city, mainly near streets. Similar to our results, they report mean noise reductions of around 4 to 6 dB(A). This was also found in the city of Milan, Italy, where recordings from 24 stations measuring from 1 January to 21 June showed an average L_DEN_ reduction of about 6 dB(A) compared to the same period in 2019 [26]. To our knowledge, there are only two studies so far reporting lower noise reductions. However, one study from Rumpler et al. (2020) [17] used the data of only one recording device. They reported a L_Aeq_ reduction in the city of Stockholm, Sweden, of 2–3 dB, depending on the weekday. The other one from the United States measured an overall reduction of about 2.6 dB(A). However, this study used the data of individual noise exposures assessed by smartwatches and headphones of 5894 participants [27]. The study defined a baseline period from 8 January to 21 February 2020, that was compared to an intervention (lockdown) period, each set depending on the state from which the data were obtained. Sound levels (expressed as equivalent continuous average exposures, normalized to 8 h exposures) were computed from Apple Watch. 

On the other hand, to date there is only one study reporting distinctly higher noise reductions ranging between 20 and 30 dB(A) at four tourist locations in Granada, Spain [28]. However, the reported reductions are based on the comparison of two recordings per location, which were also carried out at different times of day. Comparing noise levels in our study at different daytimes revealed noise reduction in this magnitude too. However, we tried not to report noise reduction that might be biased by daytime. 

Our results reveal an interesting temporal course of the overall noise decrease. The noise levels obviously started to drop as early as mid-February, however, with further decreasing levels when the lockdown measures come into force. This early decrease might be due to the situation that, in Germany, discussion about the new SARS-CoV-2 started as early as February, and voluntary restrictions, like increased use of home offices, were successively carried out by parts of the population even before the official measures were taken. 

The marked reductions in noise during lockdown measures in Germany may be caused by a substantial decrease in mobility. Dance and McIntyre (2021) [29] observed increases of about 3 db(A) at two recording stations near a highway in the period from 7 to 12 June 2020, when lockdown measures in the UK were taken back and traffic increased, accordingly. In Germany, the visibly reduced number of cars was particularly striking on the roads, e.g., less individual traffic because of closed schools, kindergartens, and home offices. In fact, the so-called traffic barometer of the Federal Highway Research Institute in Germany concludes that a marked decrease in traffic was observed on federal highways and federal roads during the lockdown period [7]. The institute compared data from one month with both the previous month’s and the same month’s data from the previous year. Across all types of motor vehicles, a decrease of 18.6% is listed for March 2020 compared to February 2020 and a decrease of 20.5% compared to March 2019 [7]. Measurements on selected highway sections in North Rhine-Westphalia by Straßen.NRW, a State Government Enterprise providing road development and maintenance services, indicate that the traffic volume the days after the lockdown reached a level of 30% compared to the same period in 2019 [30,31]. Although specific traffic mobility data for the city of Bochum are not available, one can assume that traffic reductions across the Ruhr Region are similar. 

Noise reduction due to the dramatically reduced air traffic is, however, another important effect of the measures. In Germany, air traffic has experienced a reduction of up to 60% [9]. The number of monthly take-offs has fallen from November 2019 to February 2020 from about 70,000 to less than 20,000 in April 2020 [9]. Although the city of Bochum is not located in a major flight corridor, airplanes still fly over Bochum from time to time through the proximity of Düsseldorf International Airport.

A further noise source in Germany is railway traffic. Probably, this type of source is not decisive for the noise reduction observed in our study, as restrictions were less limited. The regional railways of the main German railway provider decreased to 66% of the normal level and to 75% for the long-distance railways as of 22 April 2020 [8].

As we also analyzed our data according to temporal differences, we found interesting results. For instance, considering weekdays and weekends, the highest noise reductions occurred on weekends, strictly speaking on Sundays. This result is in line with results from Asensio et al. (2020) [14], who found a greater reduction on weekends compared to weekdays. However, during the lockdown, Sundays turned out to be somewhat louder than Saturdays. This could be explained in that we not only measured in green areas, but also in residential areas as well as major and side streets, near where people reside. As during lockdown locations like leisure parks, zoos, gyms, or playing grounds were closed, many people have gone to other somewhat unorthodox places to just be outside, spending time, exercising, or playing with children—preferably avoiding large gatherings. An even more subtle temporal resolution by analyzing the course of noise levels by hourly time of day reveals further interesting results. Except in the early morning hours, the overall pattern of noise levels pre- and during lockdown proves surprisingly similar, despite being at a lower noise level during the lockdown observation period. The converging noise levels in the early morning hours might be simply due to birdsongs in that special time of spring. With lengthening daylight, songbirds start singing loudly again, attracting mates and so on. Different to the during lockdown period, sounds of birdsongs pre-lockdown did not play a role, as it was too early in spring by that time. Birds start singing one hour before sunrise, which in Germany is at about 6:45 a.m. around 16 March, explaining the peak around 5:00 and 6:00 a.m. quite reasonably. To the best of our knowledge, no other study reported circadian rhythm of sound levels before and during the SARS-CoV-2 pandemic. Since our sound recordings are still ongoing, we will have the chance to verify this assumption fairly soon. 

### 4.1. Strengths and Limitations

As with all research, this study has several strengths and limitations. The spatially high-resolution measurements using 22 recordings from different urban land use types proved to be a strength. Differently to many other studies of urban soundscapes, our approach also includes residential areas where people most frequently reside, giving us valuable information about associations between the built and acoustic environment. The high temporal resolution as well as the long-term measurements are further strengths of this research. The systematic approach to gathering data from a wide range of urban acoustic environments while taking into account possible seasonal variances deserves mention. Additionally, the implementation of a quality management procedure, including the application of a study protocol, qualification and training of the field staff, plausibility checks of assessed sound recordings, and calculated indices increases the plausibility of the results. 

One of the limitations is that currently we do not have small-scale meteorological data. Therefore, we could not account for meteorological conditions. Especially, wind gusts directly impact the recording microphone. Furthermore, wind can cause noise on building façades as well as modify sound propagation, which changes sound pressure levels accordingly [32,33,34]. We cross-referenced meteorological wind data in Bochum to rule out the impacts of specific temporally delineated pressure systems that might have influenced wind gusts. However, these data are based on just one meteorological station. Thus, only major weather event could be taken into account, such as a strong storm between 10–11 February. Still, our results seem reasonable, since noise reductions of international studies had a similar range, and it would be unlikely that similar weather conditions were to have occurred by chance at all locations at different times. Most other studies cited here do not provide information on meteorological data. Only Bruitparif (2020) [13] mentions the exclusion of specific records of days with extreme weather events such as heavy rain and strong wind, and still reports very similar noise reduction levels. Lastly, if the reduction in noise levels were only due to lower wind speeds, then the noise reduction should actually have been more homogeneous over the different observed land uses types. 

The data collection for our automatic audio recordings started in May 2019, which is why a seasonal comparison between 2019 and 2020 is not possible. However, we do not believe that our data are mainly biased by seasonal effects for several reasons: in our data set, the seasonal differences between the time period February–April seem negligible, as for instance tree budding is still low or not yet pronounced. Furthermore, as already mentioned above, most studies to date investigating the impact of Sars-CoV-2 on noise levels have measured decreases in very similar ranges (3–5 dB), regardless of comparing the same seasonal periods between 2019 and 2020 or the weeks before and during lockdown measures. Overall, as this is an observational study, the aim is not to report causal relationships, but to describe noise levels during two very interesting periods, taking into account different land uses in an urban area.

It is tempting to assume that the measured noise reductions are mainly due to the reduced traffic volume. Although it seems reasonable to expect that the nationwide reported decreases of traffic volume are of corresponding size in Bochum, it would be desirable to have more accurate, small-scale traffic data to evaluate the effects of individual and public traffic reductions on noise levels in more detail. Our 22 recording sites already cover a wide range of urban land use types. Still more recording sites are desirable to represent the most specific places of a city available to achieve a comprehensive understanding of the urban acoustic environment.

### 4.2. Conclusions

The results presented may have implications for future urban public health measures and urban design. A reduction in traffic noise, as one main source of noise pollution [35], as well as a transformation to health-promoting urban mobility by using alternative modes of transportation, can be an effective and promising approach to mitigating the negative health effects of noise in cities. The World Health Organization estimates that one million healthy life years are lost from traffic-related noise in the western part of Europe, every year (World Health Organization 2011). Our results are in concert with other international studies cited here, showing the potential to reduce noisy and unhealthy (urban) areas through reduction of traffic volume. Urban facilities and new transportation systems [36] that provide incentives for a shift away from traditional motorized traffic modes leading to less noise and air pollution [37,38] are a promising avenue to improve urban public health. This is in line with new mobility concepts that aim to make cities more healthy, sustainable, and resilient. Overall, the experiences gleaned from the SARS-CoV-2 pandemic offer data for best practices and policies for the development of urban transportation, in that experiences of a reduced-traffic, less noisy and thus more pleasant world could be experienced firsthand by people during this time, however without a pandemic breathing down our necks in the future.

## Figures and Tables

**Figure 1 ijerph-18-04653-f001:**
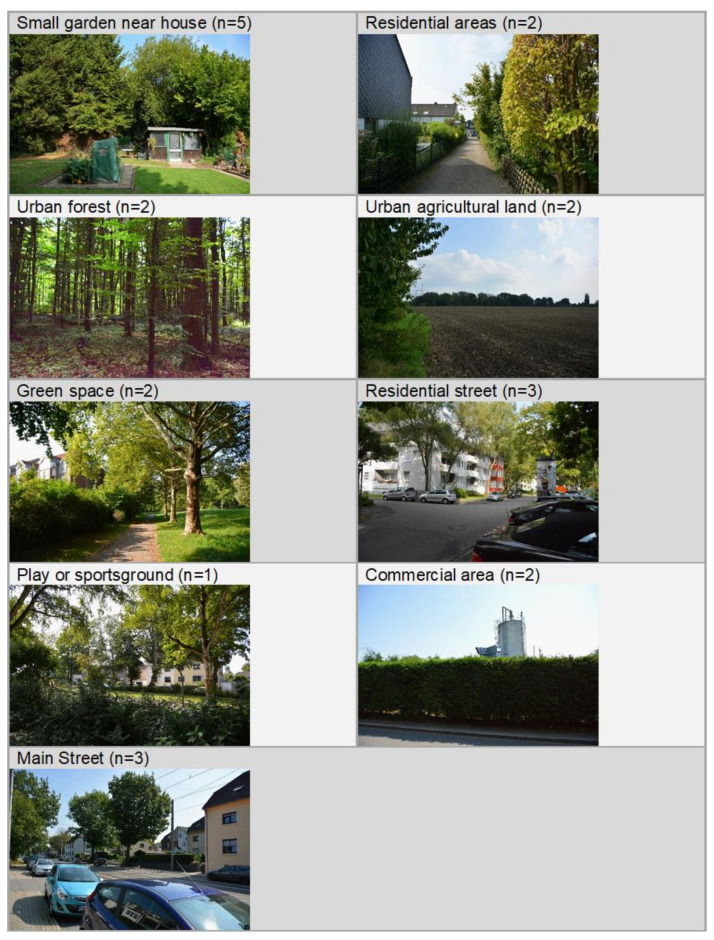
Land use categories and number (n) of recording locations by land use category (n = 22). Own photographs show examples of recorded locations in the respective land use category.

**Figure 2 ijerph-18-04653-f002:**
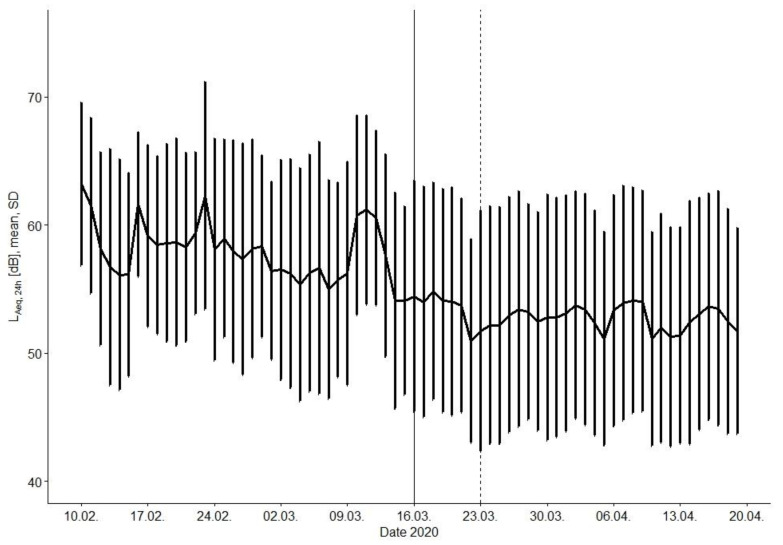
Development of noise levels, overall. Mean L_Aeq,24h_ for all 22 devices and their corresponding standard deviation (SD) are shown. Vertical full and dashed line mark the dates 16 March and 23 March according to resolved restrictions.

**Figure 3 ijerph-18-04653-f003:**
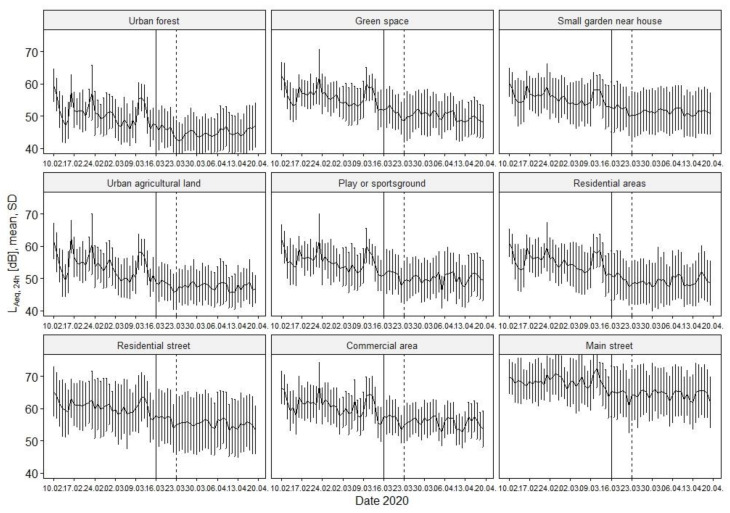
Development of noise levels by land use category. Mean L_Aeq_,_24h_ and standard deviation (SD) are shown. Vertical and dashed lines mark 16 and 23 March according to resolved restrictions (s. methods section).

**Figure 4 ijerph-18-04653-f004:**
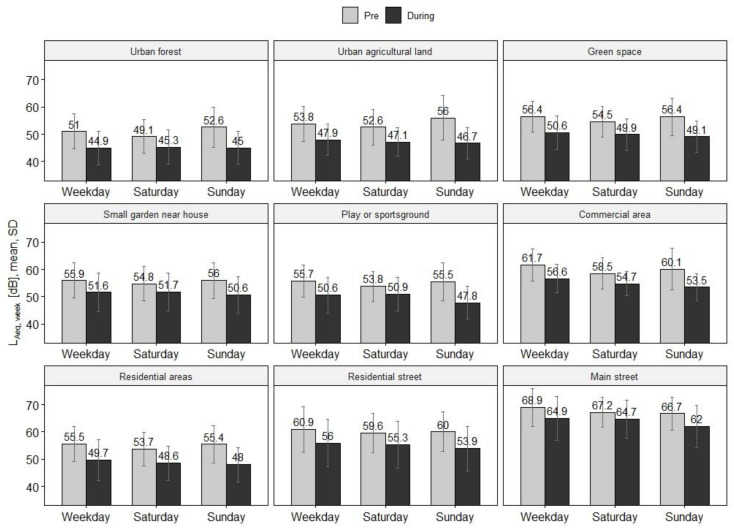
Comparison of noise levels on weekdays and weekends by land use category. Mean L_Aeq__,week_ for all *22* devices and their corresponding standard deviation (SD) are shown.

**Figure 5 ijerph-18-04653-f005:**
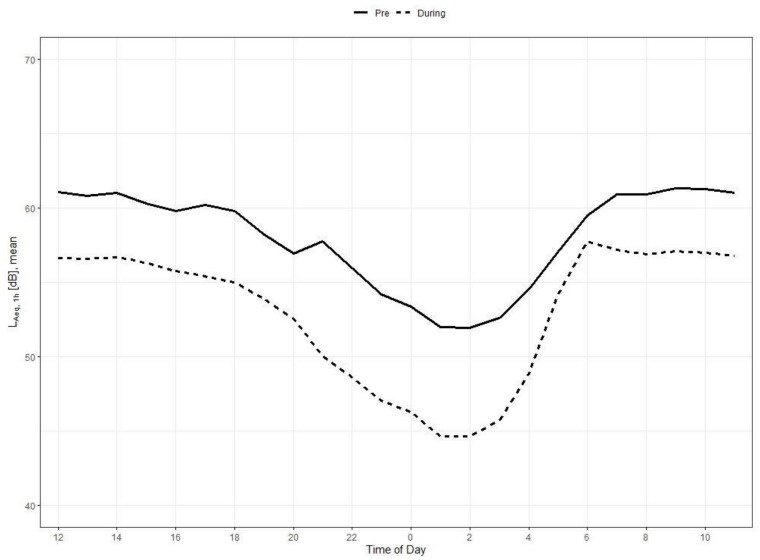
Development of hourly noise levels pre- and during lockdown, overall. Mean L_Aeq__,1h_ and standard deviation for all *22* devices. Note that time of day in this figure starts at 12:00 a.m. to better show the distinct decrease between 8:00 p.m. and 1:00 a.m.

**Figure 6 ijerph-18-04653-f006:**
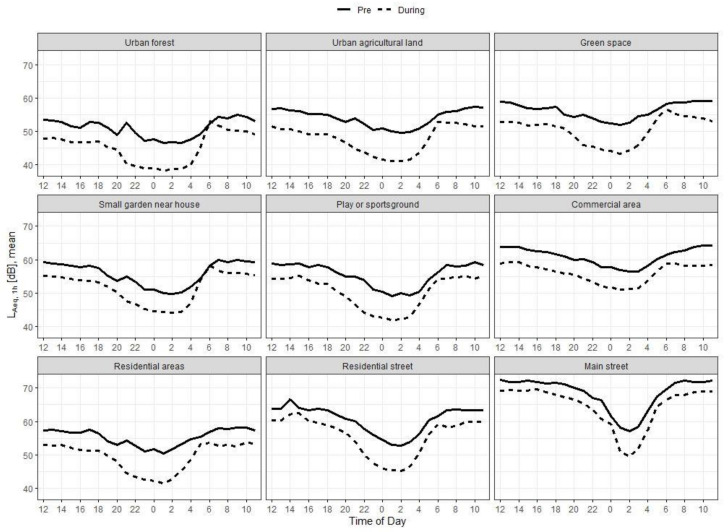
Development of hourly noise levels pre- and during lockdown by land use category. Mean L_Aeq__,1h_ and standard deviation.

**Table 1 ijerph-18-04653-t001:** Mean noise levels (LA_eq,35d_) pre- and during ($\bar{x_{p}}$ and $\bar{x_{d}}$) lockdown and noise differences, overall and by land use categories.

	Noise Levels (LAeq,35d) [dB]		
Land Use Category	$\mathrm{Pre}-\mathrm{Lockdown}\;(\bar{x_{p}})$	$\mathrm{During}\;\mathrm{Lockdown}\;(\bar{x_{d}})$	Differences (Δ)
	Mean ± SD	Mean ± SD	$\bar{x_{d}}$ $-\bar{x_{p}}$
Forest	50.9 ± 6.6	45 ± 6.2	−5.9
Green space	56.2 ± 5.8	50.3 ± 6.0	−5.9
Small garden near house	55.8 ± 6.4	51.5 ± 7.0	−4.3
Agricultural land	54.0 ± 6.8	47.6 ± 5.7	−6.4
Play or sportsground	55.4 ± 6.1	50.2 ± 6.5	−5.2
Residential area	55.2 ± 6.4	49.3 ± 7.2	−5.9
Residential street	60.6 ± 8.0	55.6 ± 8.7	−5.0
Parking lot	61.0 ± 6.3	55.9 ± 5.3	−5.1
Commercial area	68.4 ± 6.7	64.5 ± 8.0	−3.9
All	58.0 ± 8.3	52.9 ± 8.9	−5.1

**Table 2 ijerph-18-04653-t002:** Overall mean hourly noise levels and changes pre- and during lockdown.

Time 24-h Clock	Noise Levels (L_Aeq.35d_) [dB]		
	$\mathrm{Pre}-\mathrm{Lockdown}\;(\bar{x_{p}})$	$\mathrm{During}\;\mathrm{Lockdown}\;(\bar{x_{d}})$	Differences (Δ)
	Mean ± SD	Mean ± SD	$\bar{x_{d}}$ $-\bar{x_{p}}$
0	53.4 ± 8.1	46.3 ± 8.0	−7.1
1	52.0 ± 7.8	44.6 ± 6.5	−7.4
2	51.9 ± 7.5	44.6 ± 6.4	−7.3
3	52.6 ± 7.3	45.7 ± 6.6	−6.9
4	54.6 ± 7.6	48.9 ± 7.4	−5.7
5	57.1 ± 7.8	54.3 ± 7.5	−2.8
6	59.5 ± 7.2	57.8 ± 6.3	−1.7
7	60.9 ± 7.1	57.2 ± 7.1	−3.7
8	60.9 ± 7.2	56.9 ± 7.1	−4.0
9	61.3 ± 6.8	57.1 ± 7.5	−4.2
10	61.3 ± 7.1	57.0 ± 7.5	−4.3
11	61.0 ± 7.4	56.8 ± 7.5	−4.2
12	61.1 ± 7.5	56.6 ± 7.7	−4.5
13	60.8 ± 7.3	56.6 ± 7.9	−4.2
14	61.0 ± 8.2	56.7 ± 8.3	−4.3
15	60.3 ± 7.7	56.3 ± 8.6	−4.0
16	59.8 ± 7.7	55.8 ± 8.1	−4.0
17	60.2 ± 7.3	55.4 ± 7.9	−4.8
18	59.8 ± 7.4	55.0 ± 7.6	−4.8
19	58.2 ± 8.1	53.9 ± 7.9	−4.3
20	56.9 ± 8.6	52.5 ± 8.3	−4.4
21	57.7 ± 7.5	50.0 ± 8.7	−7.7
22	56.0 ± 8.1	48.6 ± 8.3	−7.4
23	54.2 ± 8.5	47.0 ± 8.0	−7.2

## Data Availability

The datasheets take up a lot of storage space, which is why we have decided not to upload them for the time being. However, the datasets generated during the current study are available from the corresponding author upon reasonable request.

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
