# Peer review of "Impact of the COVID-19 Lockdown Measures on Noise Levels in Urban Areas—A Pre/during Comparison of Long-Term Sound Pressure Measurements in the Ruhr Area, Germany"

_ijerph, 2021, doi:10.3390/ijerph18094653_

Round 1

Reviewer 1 Report

Very interesting paper and timely contribution. The literature on the noise-related changes induced by COVID lockdowns is growing, and with a reason. We need to better understand the implications of such dramatic reduction of human activities on the sound environments of our cities and regions.

The literature review picks up several examples of related studies, but others could be added to give the reader a broader context. Examples reported below.

I commend the authors for this work, I think it is well-prepared, I like the large-scale/land use approach. I only have to main comments:

  • Some more detailed description of what the administrative limitations (lockdown) would imply would be useful, for each stage (e.g., stay at home, no commercial activities, etc.): the reader would need to know what kind of human activities would be limited and potentially have a stronger impact on the acoustic environment. Adding a timeline with the sequence of events/restrictions coming into force in this German region (with references/link to government documents) would be useful.
  • Authors apparently only compared weeks before/after the lockdown; I have to say, most studies about similar topics I am reading these days are actually looking at same periods in different years (e.g., Spring 2019 vs. Spring 2020), and I tend to agree, as there is some need to control for possible seasonal variations (otherwise it could be difficult to establish clear causal connection between the lockdown and the sound levels reduction). Since data from 2019 is available, could the authors replicate the same analysis they have already done for this layer of temporal information?

Please use these just as a starting point and mention in Intro or Discussion:

Manzano, Jerónimo Vida, Pastor, José Antonio Almagro, Quesada, Rafael García, Aletta, Francesco, Oberman, Tin, Mitchell, Andrew and Kang, Jian. "The “sound of silence” in Granada during the COVID-19 lockdown" Noise Mapping, vol. 8, no. 1, 2021, pp. 16-31. https://doi.org/10.1515/noise-2021-0002

Dance, Stephen and McIntyre, Lindsay. "The Quiet Project – UK Acoustic Community’s response to COVID19 during the easing of lockdown" Noise Mapping, vol. 8, no. 1, 2021, pp. 32-40. https://doi.org/10.1515/noise-2021-0003

Zambon, Giovanni, Confalonieri, Chiara, Angelini, Fabio and Benocci, Roberto. "Effects of COVID-19 outbreak on the sound environment of the city of Milan, Italy" Noise Mapping, vol. 8, no. 1, 2021, pp. 116-128. https://doi.org/10.1515/noise-2021-0009

Author Response

Dear reviewer

Thank you very much for giving us the opportunity to address and include your comments in this manuscript for re-submission. Thank you for your work and valuable contributions to improve the manuscript.

Please find in the following pages your comments and our responses. Note that we refer in our comments to the line numbers in the current revised Word file, in which changes have been highlighted.

Additional changes are listed in the end.

The newly created figures are inserted in this document, but we kindly ask the editors to insert them as an appendix in the manuscript according to the formatting. 

Best regards,

Jonas Hornberg

Reviewer 2 Report

Ijerph 1193312 Review

This is a nice addition to the literature. The paper has a few minor issues that need to be addressed

In the introduction the authors may want to consider that there is also additional literature that looks at the effects of COVID on keynote sounds (eg. Parker & Spennemann 2020), but they the paper will only focus on general noise levels, primarily traffic generated.

In France, the US and Span  residents during lockdown engaged in a special clapping event to celebrate the efforts of frontline workers. This should be referenced and discussed whether tis occurred in Bochum or not.

Can the authors explain why there is a discernible downward trajectory in noise levels in the pre-lockdown phase. To what extent did COVID-19 concerns already influence behaviour? Looking that the graphs, there appears to be a uniform downward trend interpunctuated by some spikes in the pre-lockdown period

What explains the quite uniform and distinct peak 10-12 March?

More generally, it might be advantageous compare the 2020 lockdown period with the same period in 2019 (see Spennemann & Parker 2020).

Minor quibbles

Line 40 needs reference

Line 106 ff explain to what extent travel to work was still permissible during lock down as this will influence commuting

Line 333 Do you have traffic count data for Bochum for the period under discussion? You say ‘no’ , but did you check for traffic light actuation data?  Usually they do exist… I would be surprised if Bochum has no traffic management system that relies on quantitative date from loop counters or traffic cams…I would suggest to add endnote or explanation what avenues have been attempted to source the data…

Suggestions

Include a map with the locations of the sound monitoring stations

Fig 1 Make the images a bit bigger so that one can actually see your classification

See also:

Parker, M., & Spennemann, D. H. R. (2020). Anthropause on audio: the effects of COVID-19 pandemic on church bell ringing in New South Wales (Australia). Journal of the Acoustical Society of America, 148(5), 1–5. doi:10.1121/10.0002451

Spennemann, D. H. R., & Parker, M. (2020). Hitting the ‘Pause’ Button: what does COVID tell us about the future of heritage sounds? Noise Mapping, 7, 265–275. doi:10.1515/noise-2020-0022

Author Response

(The authors gave the same response as above.)

Round 2

Reviewer 1 Report

I think the authors have addressed well all the points, I am happy to endorse the manuscript.